# Structural basis for the self-recognition of sDSCAM in Chelicerata

Jie Cheng[1,8], Yamei Yu [2,8], Xingyu Wang[2,8], Xi Zheng[3,4,8], Ting Liu[2], Daojun Hu[2], Yongfeng Jin [5], Ying Lai [1], Tian-Min Fu [6,7] & Qiang Chen [2] ✉

To create a functional neural circuit, neurons develop a molecular identity to discriminate self from non-self. The invertebrate Dscam family and vertebrate Pcdh family are implicated in determining synaptic specificity. Recently identified in Chelicerata, a shortened Dscam (sDscam) has been shown to resemble the isoform-generating characters of both Dscam and Pcdh and represent an evolutionary transition. Here we presented the molecular details of sDscam self-recognition via both *trans* and *cis* interactions using X-ray crystallographic data and functional assays. Based on our results, we proposed a molecular zipper model for the assemblies of sDscam to mediate cell-cell recognition. In this model, sDscam utilized FNIII domain to form side-by-side interactions with neighboring molecules in the same cell while established hand-in-hand interactions via Ig1 domain with molecules from another cell around. Together, our study provided a framework for understanding the assembly, recognition, and evolution of sDscam.

The complexity of eukaryotic nervous system is established via neuronal cell interactions with vast diversity and specificity. During development, neurons need to discriminate self from non-self to establish appropriate connections. The neuronal wiring and self-avoidance rely on the extraordinary recognition diversity of cell surface molecules, such as the *Drosophila* Down syndrome cell adhesion molecules (Dscams)[1,2] and the mammalian clustered protocadherins (cPcdhs)[3]. *Drosophila Dscam1* encodes 38,016 distinct isoforms via mutually exclusive RNA splicing[1], while human clustered *Pcdh* generates 52 isoforms using alternative promoters[4]. Recently, a shortened *Dscam* (sDscam) gene family with tandemly arrayed 5′ cassettes in Chelicerata species has been identified, which encodes ~50–100 isoforms[5,6]. With high sequence similarities to *Drosophila* Dscam1 and a striking gene organizational resemblance to the 5′ variable region of vertebrate clustered Pcdhs, Chelicerata sDscam expanded its isoform diversity via a combination of alternative promoter and RNA splicing

selections, representing remarkable functional convergence of invertebrate Dscams and vertebrate cPcdhs[5].

Both invertebrate Dscam and vertebrate cPcdh isoforms exhibit isoform-specific *trans* homophilic interactions to generate avoidance signals[2,3,7]. *Drosophila* Dscam1 uses three variable Ig domains (Ig2, Ig3, and Ig7) to perform homophilic binding while mammalian cPcdh's homophilic recognition is mediated by N-terminal EC1–EC4 Domains[8–10]. The *cPcdh* genes are organized into three tandem clusters, *Pcdhα*, *Pcdhβ*, and *Pcdhγ*[4]. Chelicerata sDscams could be classified into two subfamilies, sDscamα and sDscamβ, containing one and two variable Ig domains, respectively[5]. Different from the *Drosophila* Dscam1 that contains 10 immunoglobulin (Ig) domains and 6 fibronectin type III (FNIII) repeats, Chelicerata shortened sDscam only has 3 Ig domains and 3 FNIII domains in its extracellular part, corresponding to Ig7-Ig9 and FNIII1, 2, 5 of Dscam1, respectively[5] (Fig. 1). Chelicerata sDscams also exhibited isoform-specific *trans* homophilic

[1]National Clinical Research Center for Geriatrics, West China Hospital, State Key Laboratory of Biotherapy and Collaborative Innovation Center of Biotherapy, Sichuan University, 610041 Chengdu, China. [2]State Key Laboratory of Biotherapy and Cancer Center, West China Hospital, Sichuan University, and Collaborative Innovation Center of Biotherapy, 610041 Chengdu, China. [3]Department of Thoracic Surgery, West China Hospital, Sichuan University, 610041 Chengdu, China. [4]Lung Cancer Center, West China Hospital, Sichuan University, 611135 Chengdu, China. [5]MOE Laboratory of Biosystems Homeostasis & Protection and Innovation Center for Cell Signaling Network, College of Life Sciences, Zhejiang University, 310058 Hangzhou, Zhejiang, China. [6]Department of Biological Chemistry and Pharmacology, The Ohio State University, Columbus, OH 43210, USA. [7]The Ohio State University Comprehensive Cancer Center, Columbus, OH 43210, USA. [8]These authors contributed equally: Jie Cheng, Yamei Yu, Xingyu Wang, Xi Zheng. ✉e-mail: qiang_chen@scu.edu.cn

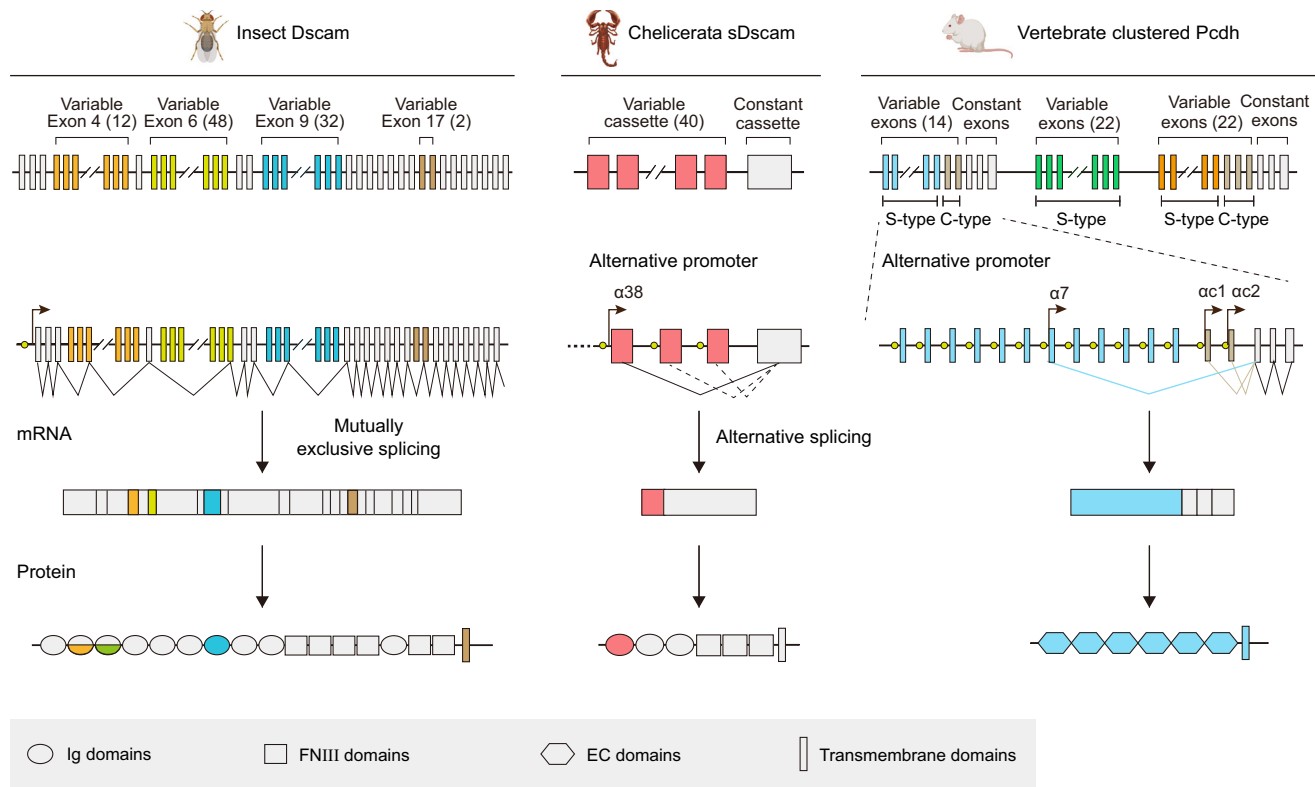

**Fig. 1 | Schematic depiction of *Dscam*, *sDscam* and *cPcdh* gene organization and isoform diversity.** *Drosophila melanogaster Dscam*, *Mesobuthus martensii sDscam* and mouse *cPcdh* have been chosen as samples. The variable regions are indicated by different colors, while the common regions are colored in gray. The promoters are represented as yellow filled circles. *Dscam* applies mutually exclusive RNA splicing to generate tens of thousands of isoforms. For *cPcdh*, each variable exon is preceded by a promoter and only the cap-proximal exon engages in the splicing with the first constant exon. The shorten *sDscam* uses both alternative promoter and RNA splicing to generate its isoform diversity. For clarity, only the α subfamilies are shown for *sDscam* and *cPcdh*. More details of *sDscam* isoforms are shown in Supplementary Fig. 1. The species icons are created with MedPeer (www.medpeer.cn).

interactions[11]. Since the generation of a large number of isoforms and the isoform-specific interactions are two hallmarks of Dscam/cPcdh-mediated neuron recognition and self-avoidance, this shortened sDscam likely plays a similar functional role as Dscam and cPcdh.

Given the huge discrepancy in isoform diversity between Dscam (38,016 isoforms in *Drosophila*) and cPcdh (52 isoforms in human), there is a big evolutionary gap between insects and vertebrates. Chelicerata sDscam, which resembles the isoform-generating features of both invertebrate Dscam and vertebrate cPcdh, provides us with a unique opportunity to get a glimpse of the evolutionary transitions.

Here, we present 12 crystal structures of sDscam fragments Ig1, Ig1-2, Ig1-3, FNIII1, FNIII2, FNIII3, FNIII2-3, and FNIII1-3, establishing the binding behavior of the *trans* (cell-to-cell) and *cis* (the same cell surface) binding modes for Chelicerata sDscams. Crystal structures and cell aggregation assays demonstrated that both α and β subfamilies of sDscam used Ig1 for the isoform-specific *trans* recognition in a hand-in-hand manner. Furthermore, the *cis* interactions were established by the three FNIII domains of sDscam, which assemble into a cross fold with a kink between FNIII2 and FNIII3. Based on the limited isoform diversity and the *trans* and *cis* binding modes of sDscam, we proposed a zipper-like model for the assemblies formed by the entire sDscam ectodomain between cells. This study provided insights into sDscam-mediated cell discrimination and deepened our understandings of the evolutionary scenario of cell recognition molecule diversity.

## Results

### Structure determination of sDscam

A shortened sDscam in Chelicerata combines alternative promoter and RNA splicing to generate diverse isoforms, resembling both insect Dscams and vertebrate Pcdhs[5] (Fig. 1). In chelicerate *Mesobuthus martensii*, *sDscam* gene family contains tandemly arrayed cassettes, which comprise two or four exons encoding a single or two variable Ig domains, respectively (Supplementary Fig. 1), and thus can be classified as sDscam α and β subfamilies. In the current study, we determined the crystal structures of sDscam from both α and β subfamilies to provide molecular details for its *trans* and *cis* recognitions.

A recent study showed that the *trans* homophilic interactions of Chelicerata sDscam were mediated via its N-terminal Ig1 domain[11]. To elucidate the molecular details of *trans* interactions of sDscam, we determined the crystal structures of Ig1 from *Mesobuthus martensii* sDscam isoforms α1, α7, and β6v2. To validate Ig2 and Ig3 are not involved in the *trans* homodimerization of sDscam, we further determined the crystal structures of the fragments Ig1-2 (isoform β2v6) and Ig1-3 (isoform α25) from *M. martensii*. We also determined the crystal structures of sDscam Ig1 from another chelicerate species *Limulus polyphemus* (isoforms α7 and β3v7) to provide insights into the conservation of sDscam *trans* recognition within Chelicerata. These crystal structures of Ig fragments were determined at 1.3–3.1 Å resolutions (Supplementary Table 1 and Supplementary Table 2).

It has been speculated that Chelicerata sDscam could form *cis* multimers via membrane proximal FNIII domains[11]. To analyze the *cis* interactions of sDscam, we determined the crystal structures of different FNIII fragments of *M. martensii* sDscam: FNIII1, FNIII2, and FNIII3 domains of isoform α7, and FNIII2-3 and FNIII1-3 fragments of isoform β2v6. These crystal structures of FNIII fragments were determined at 1.4 ~ 3.2 Å resolutions (Supplementary Tables 1 and 2).

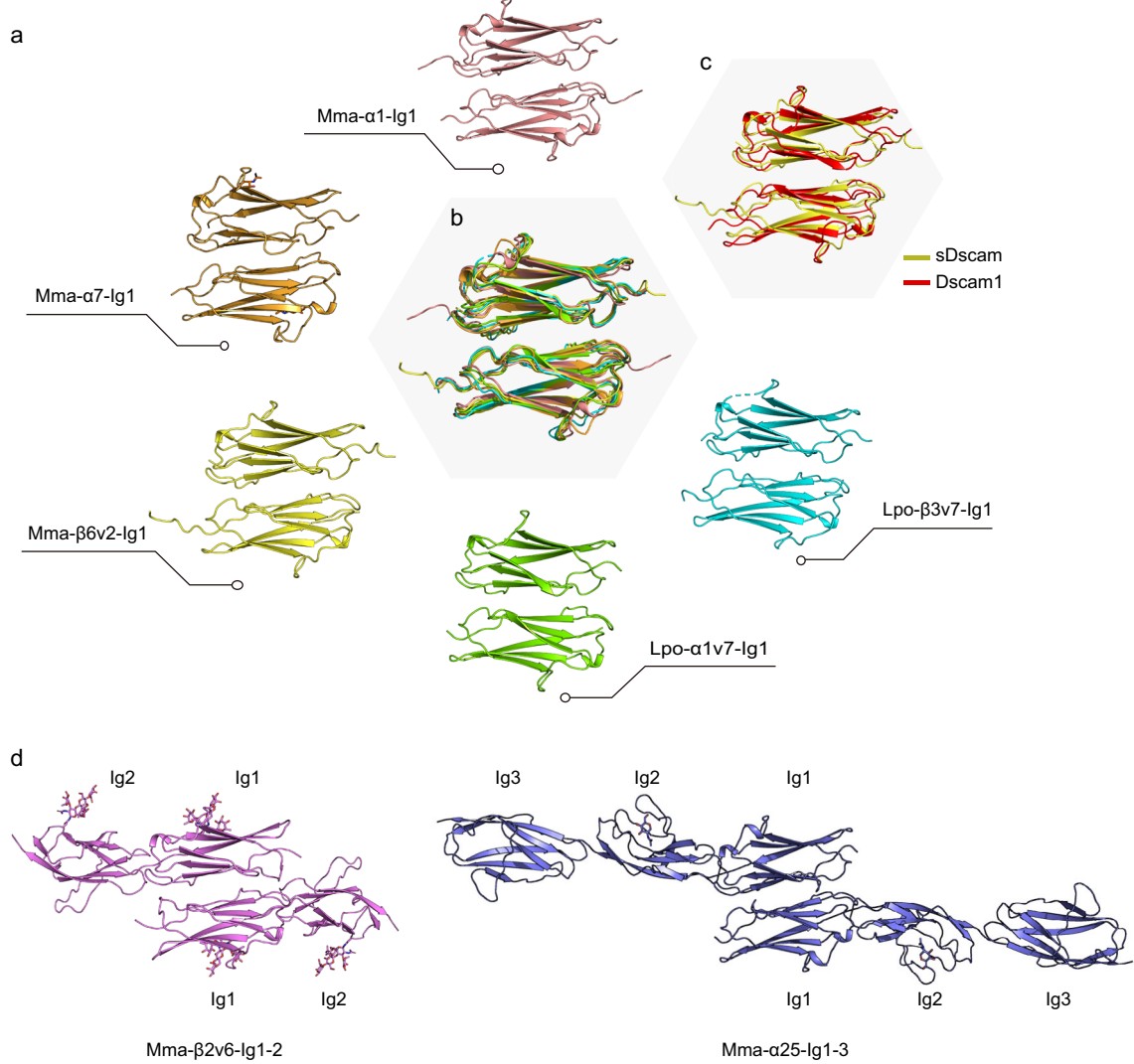

**Fig. 2 | The homodimer of sDscam Ig1. a** The crystal structures of Ig1 homodimers from *M. martensii* and *L. polyphemus*. Glycans are shown in stick representation. **b** Superposition of sDscam Ig1 homodimers. **c** Superposition of the homodimers of sDscam Ig1 and Dscam1 Ig7 (PDB code 4WVR [https://www1.rcsb.org/structure/4WVR]). **d** The crystal structures of *M. martensii* Ig1-2 and Ig1-3 homodimers.

## Structural basis for the *trans* homodimerization of sDscam

The crystal structures of sDscam Ig1 from both *M. martensii* and *L. polyphemus* revealed a common assembly of antiparallel homodimer for both α and β subfamilies of sDscam (Fig. 2a). In addition, all these Ig1 homodimers resemble each other well with root mean square deviation (RMSD) values of 1.0–1.6 Å, indicating a shared mechanism of sDscam *trans* recognition within Chelicerata (Fig. 2b). As the homolog of *Drosophila* Dscam1 Ig7, sDscam Ig1 adopted a very similar conformation as *Drosophila* Dscam1 Ig7 homodimer (Fig. 2c), highlighting a conserved mechanism for the *trans* recognition in Insecta Dscam and Chelicerata sDscam. The fragments Ig1-2 (isoform β2v6) and Ig1-3 (isoform α25) both formed *trans* homophilic dimers exclusively through interactions mediated by the Ig1 domain (Fig. 2d). The Ig2 and Ig3 domains in Ig1-2 and Ig1-3 linearly followed after Ig1 domain without participating *trans* interactions.

The Ig fold is featured by a sandwich of two β-sheets, denoted as ABED face and CFG face, respectively. A previous study has demonstrated that *Drosophila* Dscam1 Ig7 use its ABED face to form a homophilic dimer through a charge complementary mechanism to ensure an antiparallel binding mode[8]. Similarly, sDscam Ig1 also employs the ABED face to form an antiparallel homodimer (Fig. 2). However, among the seven Ig1 structures reported here, the

complementary electrostatic potential surface pattern (negative in one end, neutral in the middle, and positive in the other end) has been only observed in the three β isoforms, but not in any one of the four α isoforms (Fig. 3a and Supplementary Fig. 2). In addition, the electrostatic potential arrangement of sDscam Ig1 ABED face was opposite to that of *Drosophila* Dscam1 Ig7 (Fig. 3b).

More interestingly, in some Ig1 crystal structures (*M. martensii* α1, α7, and *L. polyphemus* α7), the two protomers within a homodimer were related via a 2-fold symmetry axis, namely, these two protomers had identical conformations. This implied that the homophilic *trans* interactions of Ig1 were also symmetry-related.

To reveal the principles of Ig1 interactions, we analyzed all the interfaces of Ig1 homodimers. Structural analysis showed that B and E strands constituted the core of ABED face and dominated the *trans* homophilic interactions of sDscam Ig1. The residue preceding the conserved B-strand cystine always interacts with its counterpart of the other protomer. Electrostatic and shape complementarity had been observed in all Ig1 dimer interfaces (Fig. 3c). The electrostatic repulsion and/or shape non-complementarity may avoid the binding between different sDscam isoforms, resulting in isoform-specific interactions. Consistently, the sequence alignment of 40 isoforms of *M. martensii* sDscamα subfamily showed low conservation for the ABED face, but

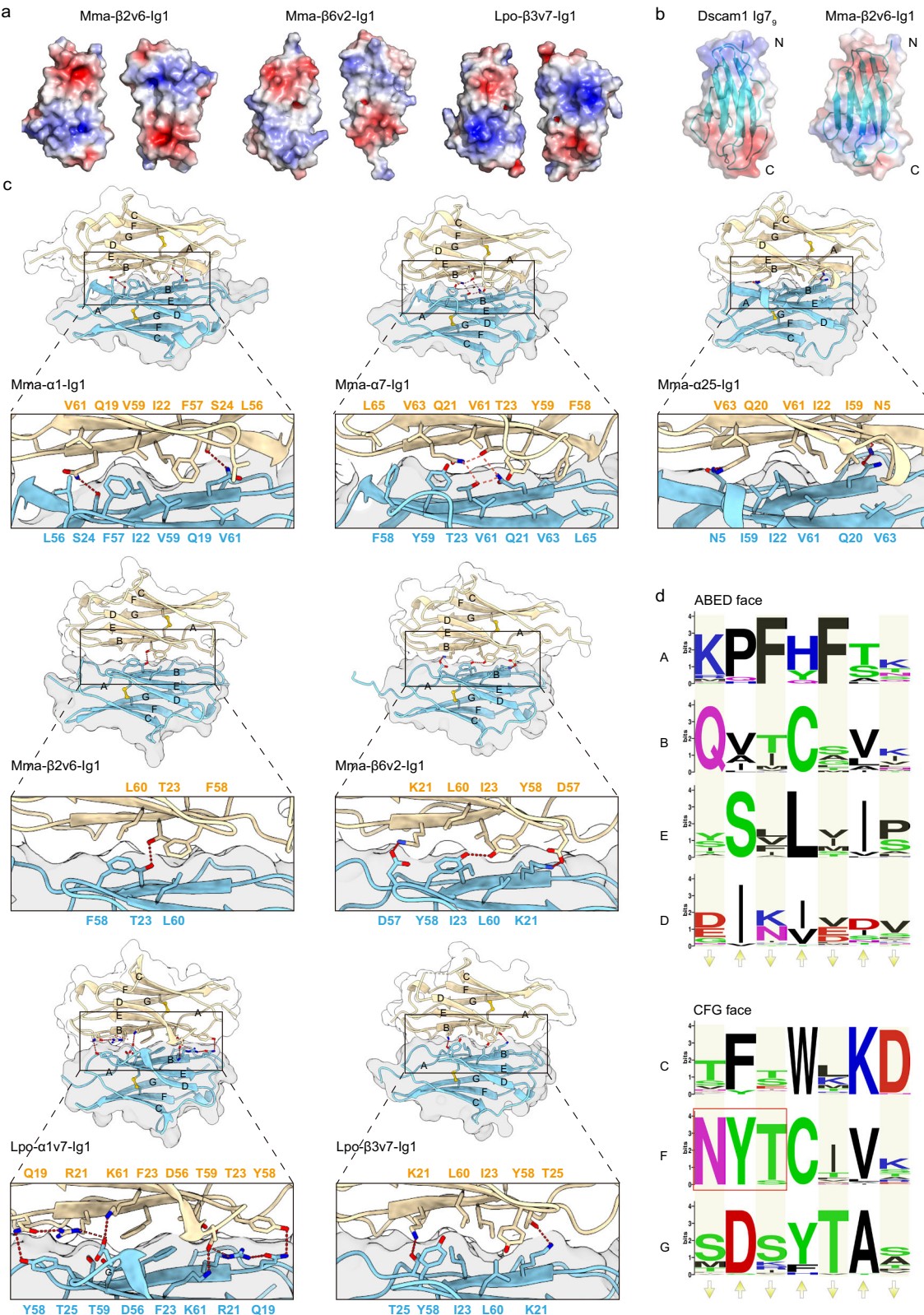

**Fig. 3 | The interface of Ig1 homodimer. a** Open-book view of the electrostatic surface potential of the homodimers of Ig1 isoforms. Blue and red (±5 kT/e) indicate the positively and negatively charged areas, respectively. **b** The electrostatic surface potential patterns of Dscam1 Ig7 (PDB code 4X5L [https://www1.rcsb.org/structure/4X5L]) and sDscam Ig1 display opposite orientations. The N-terminus and C-terminus of the Ig domain are indicated. **c** The Ig1 homodimer interfaces are viewed down the two-fold symmetry axis. The two protomers within an Ig1 homodimer are colored in wheat and cyan, respectively. The details of the interface are shown in the enlarged part. **d** Sequence logo representation of the conservation of Ig1 isoforms from *M. martensii* sDscam. A total of 40 isoforms of the sDscam α subfamily are used for comparison. The up arrows and down arrows indicate the inner-facing and outer-facing residues, respectively. The outer-facing residues are shaded. The constant *N*-glycosylation motif (NXT/S) is indicated by a red box.

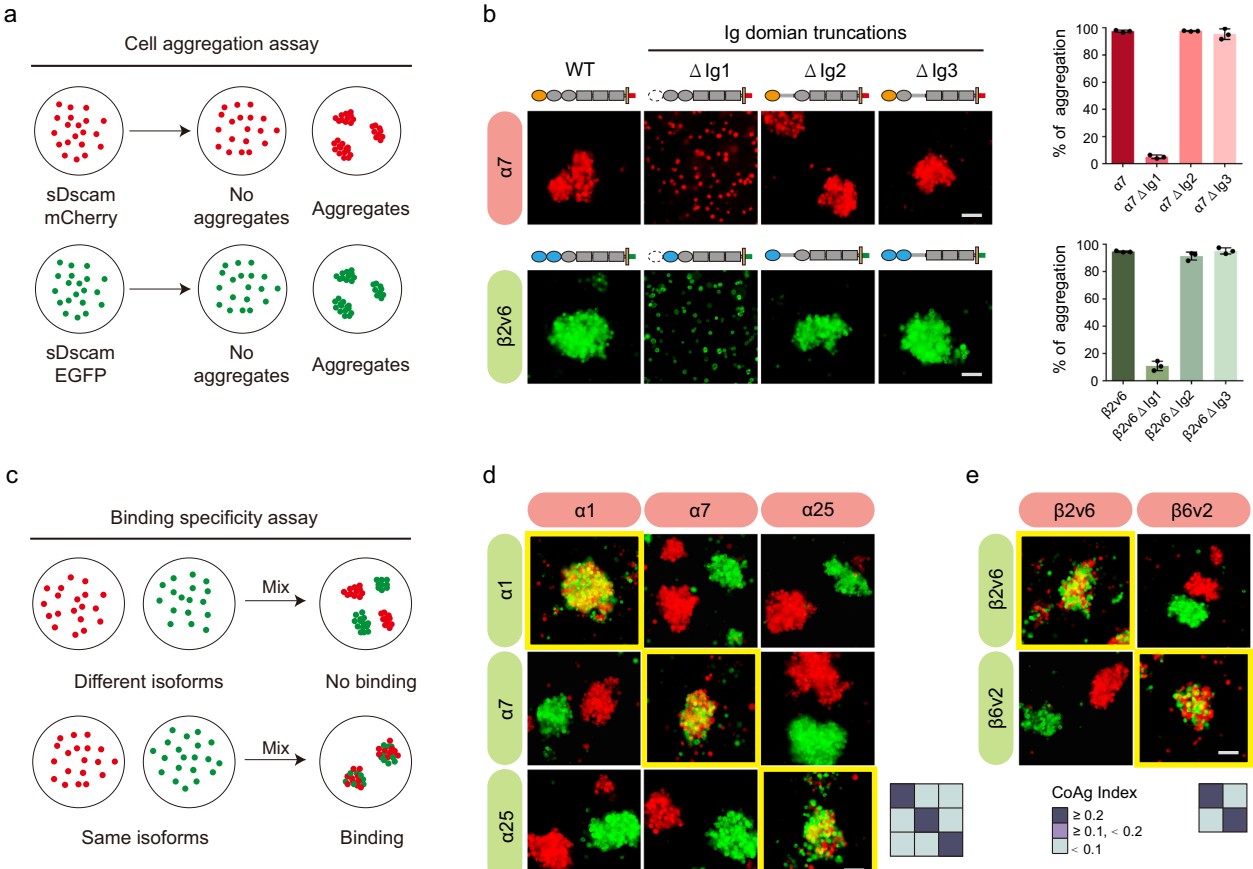

**Fig. 4 | The Ig1 mediates the isoform-specific homophilic interactions of sDscam. a** Schematic diagram of the cell aggregation assay. The mCherry- or EGFP-tagged sDscam proteins are expressed on Sf9 cells to their ability to induce cell aggregation. **b** The wild type or truncation sDscam proteins are expressed on Sf9 cells to assay for their ability to form cell aggregates. The ratios of cell aggregation are quantified. Values are means + SD ($n = 3$ independent experiments). **c** Schematic diagram of the binding specificity assay. Cells expressing mCherry- or EGFP-tagged sDscam isoforms are mixed and assayed for homophilic or

heterophilic binding. The *trans* interaction between cells expressing sDscam isoforms with different tags is indicated by mixed red-green coaggregation, while the negative results are indicated by red-green segregation. **d** Pairwise combinations within three sDscamα isoforms are assayed for their binding specificity. **e** Pairwise combinations within two sDscamβ isoforms are assayed for their binding specificity. Scale bar, 100 μm. The coaggregation (CoAg) index is quantified and illustrated as a heat map. Three independently performed experiments with similar results. Source data are provided as a Source Data file.

high conservation of CFG face (Fig. 3d). In particular, the β strands of an Ig domain have alternate distribution of inner-facing and outer-facing residues, and we observed a much higher variability of the outer-facing residues that mediate Ig1 dimerization, especially Ig1 E strand (Fig. 3d). Moreover, all the Ig1 isoforms without exception have a potential N-linked glycosylation site (NXS/T) located at the F strand (Fig. 3d). This glycosylation prevents the CFG face from establishing homophilic interactions and helps define the interface of the sDscam homodimer.

**Isoform-specific inter-cell recognition of sDscam**

To validate the functional significance of the *trans* recognitions of sDscam observed in the crystal structures, we measured the ratio of aggregated cells mediated by wild type and truncated sDscams over-expressed in cells using a cell aggregation assay (Fig. 4a). For both α and β subfamilies, wild type sDscams lead to cell aggregation, while truncation of Ig1 domain, but not other Ig domains, completely abolished cell aggregation (Fig. 4b). This finding supported the functional significance of Ig1 in mediating the *trans* recognition of sDscam, which was consistent with our structural analysis (Fig. 2).

To validate the isoform-specific interaction mediated by sDscam, we examined cell aggregation using cells expressing different isoforms of sDscam. For each isoform of *M. martensii* sDscam, we fused mCherry and EGFP to its C-terminus respectively and evaluated the

binding specificity by pairwise isoform combinations (Fig. 4c). Cells expressing the same isoforms of sDscam formed aggregation while cells with different isoforms of sDscam were separated from each other (Fig. 4d, e). These data showed that all the tested isoforms exhibited a strict *trans* homophilic specificity.

**Detection of the *cis* polymerization of sDscam on living cell surface**

A recent study suggested that Chelicerata sDscam utilized its membrane proximal FNIII domains to form *cis* interactions[11]. To identify the specific FNIII domain responsible for *cis* interactions, we developed a living cell-based assay. In this assay, the intracellular portion of the receptor tyrosine kinase cKIT and a Flag tag were fused to the FNIII domains of sDscam to make a sDscam-cKIT chimera (Fig. 5a). As monomeric cKIT cannot phosphorylate itself, the neighboring cKIT molecules phosphorylate each other when they assemble into oligomers. The phosphorylation intensity is correlated to quantities of oligomerized molecules. By monitoring the cKIT tyrosine phosphorylation through anti-phospho-tyrosine immunoblotting in SF9 cells, we can quantify the number of oligomerized sDscam molecules.

We found that FNIII2 alone but not FNIII1 or FNIII3 could promote cKIT tyrosine phosphorylation (Fig. 5b), suggesting the essential role of FNIII2 domain in mediating sDscam *cis* interactions. Furthermore,

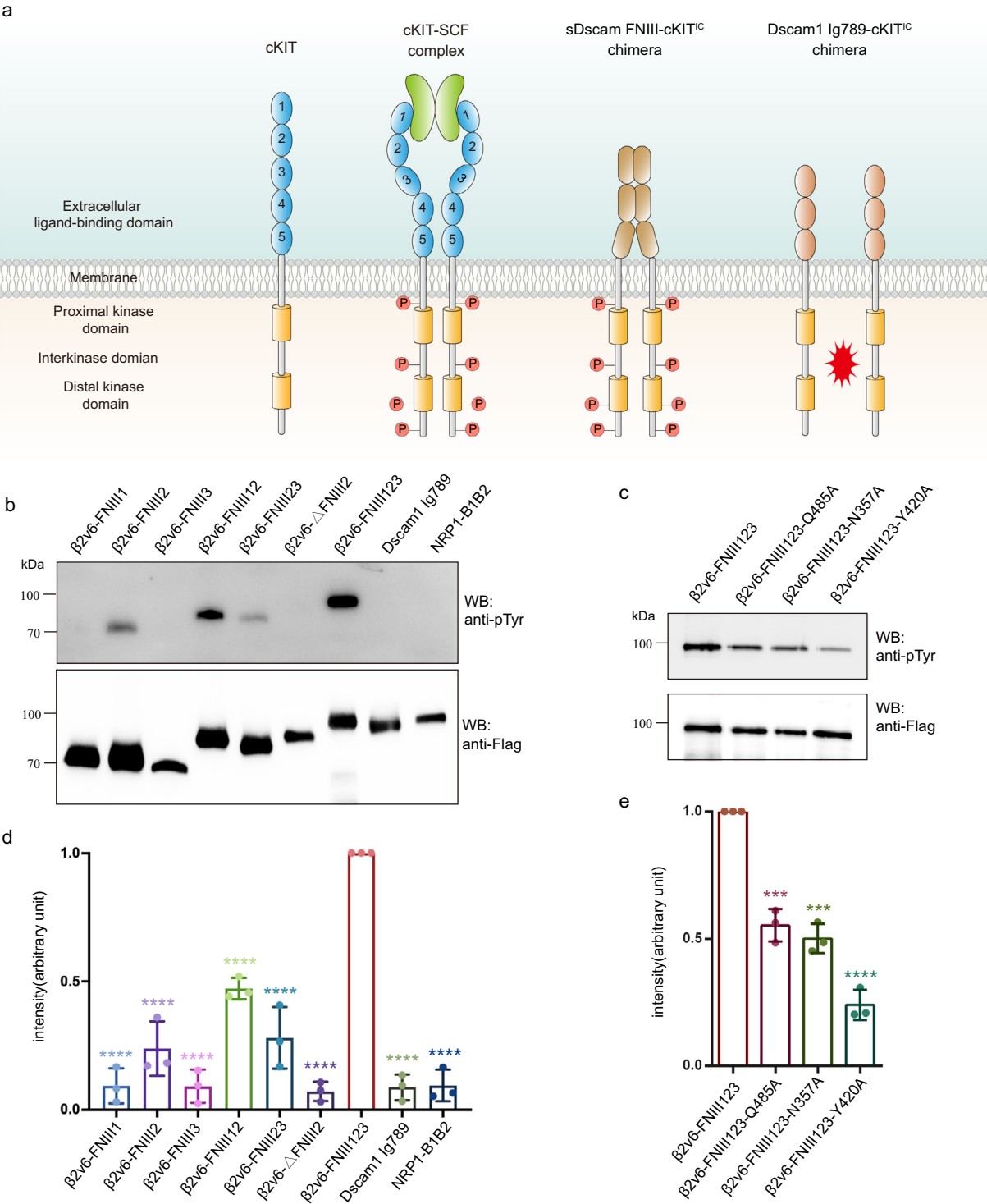

**Fig. 5 | Detection of sDscam *cis* interactions on living cell surface. a** Schematic diagram of the phosphorylation dimerization assay. The sDscam-cKIT chimera are expressed on Sf9 cell surface. If dimerization occurs, the intracellular portion of cKIT gets phosphorylated, and the phosphorylation signals are then monitored through anti-phospho-tyrosine immunoblotting. **b** Immunoblotting analysis of cKIT tyrosine phosphorylation. The same samples are detected through anti-phospho-tyrosine and anti-Flag immunoblotting to monitor the phosphorylation protein and the total protein, respectively. Upper panel, anti-phospho-tyrosine immunoblotting. Lower panel, anti-Flag immunoblotting. The Ig7-9 fragment of *Drosophila* Dscam1 and a nonrelative protein (human NRP1 B1B2 domain) are used as negative controls. For ΔFNIII2 construct, the intermediate FNIII2 domain is replaced by a 15-residue GGS linker. **c** Verification of the *cis*-dimer interfaces by mutagenesis study. The FNIII1-FNIII2 interface mutants N357A and Q485A, and the FNIII2-FNIII2 interface mutant Y420A are studied by cKIT tyrosine phosphorylation assay. **d, e** Quantification of the cKIT tyrosine phosphorylation. The intensities of the bands in **b** and **c** are calculated. The phosphorylation signals (anti-phospho-tyrosine immunoblotting intensities) are normalized by the protein level of each sample (anti-Flag immunoblotting intensities). The value of FNIII1-3 is set as 100%. Values are means + SD ($n = 3$ independent experiments). ***$P = 3$e-04 (Q485A), ***$P = 1$e-04 (N357A), ****$P < 0.0001$, calculated by two-tailed Student's *t* test, the exact $P$ values below 1e-04 are not available. Source data are provided as a Source Data file.

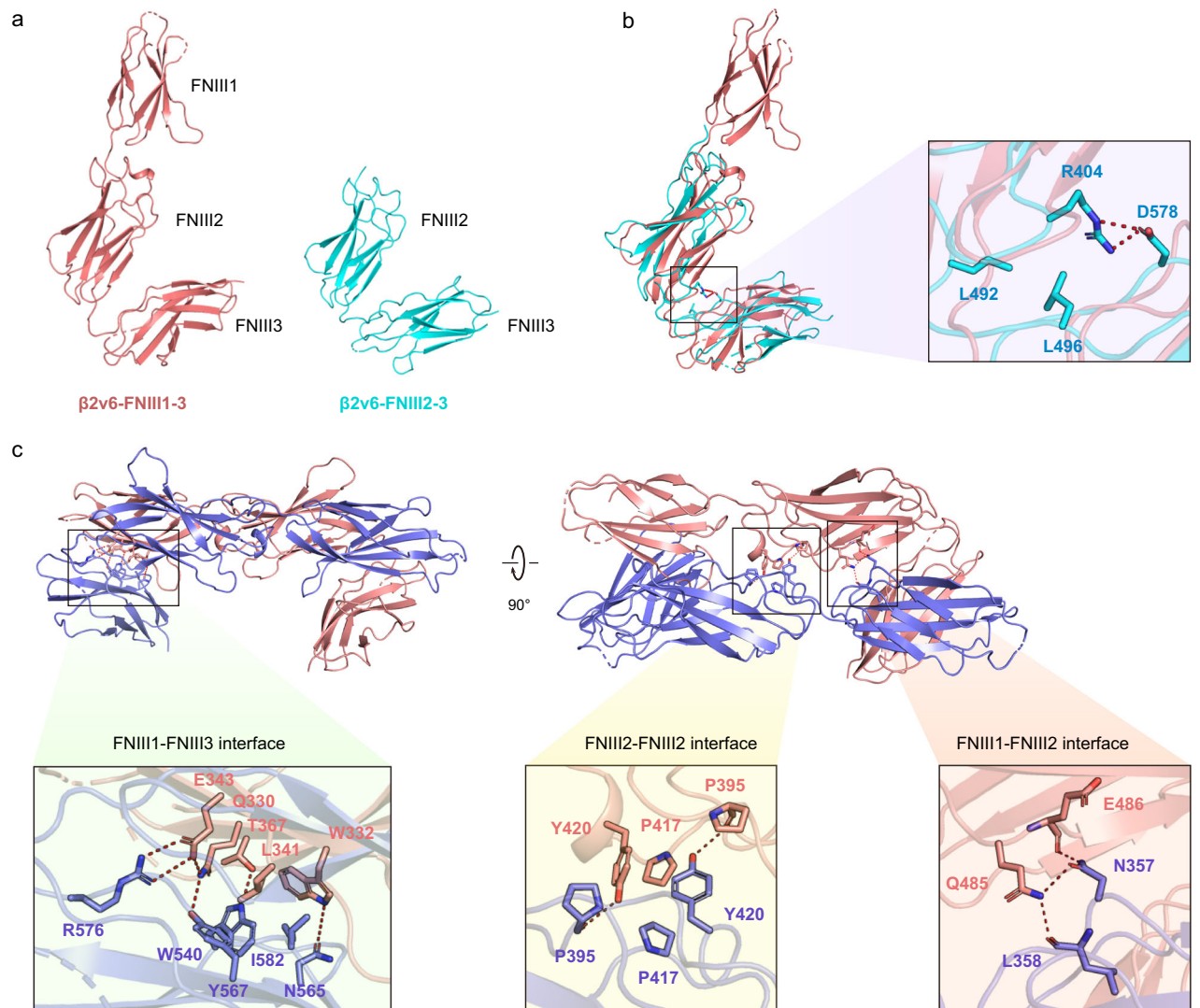

**Fig. 6 | The homodimer of sDscam FNIII domains. a** The three FNIII domains of sDscam adopt an L-shaped configuration. A kink between FNIII2 and FNIII3 are observed in both FNIII1-3 (red) and FNIII2-3 (cyan) structures. **b** Superposition of the crystal structures of sDscam FNIII1-3 (red) and FNIII2-3 (cyan). The residues involved in the inter-domain interactions between FNIII2 and FNIII3 are shown as stick and labeled. **c** The homodimer of sDscam FNIII domains adopt a cross configuration. The details of three interfaces are shown in the enlarged part.

significantly stronger cKIT tyrosine phosphorylation signal was observed in FNIII1-3 fragment, compared to FNIII2 alone, FNIII12, or FNIII2-3 fragments (Fig. 5d), indicating that the three FNIII domains cooperate to establish an optimal *cis* recognition of sDscam.

**Structural basis for the *cis* homodimerization of sDscam**
To elucidate the structural basis for the *cis* interactions of sDscam, we determined the crystal structures of different FNIII fragments of *M. martensii* sDscam: FNIII1, FNIII2 and FNIII3 domains of isoform α7, and FNIII2-3 and FNIII1-3 fragments of isoform β2v6. In contrast to the elongated conformation of sDscam Ig1-3, FNIII1-3 adopted an "L" shape resulting from a kink between the FNIII2 and FNIII3 domains (Fig. 6a). Similarly, the kink between FNIII2 and FNIII3 was also observed in the crystal structure of FNIII2-3 fragment (Fig. 6a). Notably, there were seven FNIII2-3 molecules per asymmetric unit, which all adopted a very similar kink conformation with root-mean-square deviation (RMSD) of 0.727 Å over 198 Cα atoms (Supplementary Fig. 3). Thus, this kink conformation may reflect a physiological geometric assembly. The inter-domain interface of the kink contains a salt bridge formed by R404 and D578, and the hydrophobic interaction mediated by L492 and L496 (Fig. 6b).

In contrast to the previously proposed parallel model[11], two L-shaped sDscam FNIII1-3 molecules formed a homodimer with a cross configuration (Fig. 6c). The dimer interface buries 2109 Å² of surface area, which is composed of one FNIII2-FNIII2 interface, two FNIII1-FNIII2 interfaces and two FNIII1-FNIII3 interfaces (Fig. 6c). Consistent with our cellular data (Fig. 5), FNIII2 played a pivotal role in sDscam *cis* recognition as it formed intermolecular interactions with FNIII1 and FNIII2 of the other protomer and intramolecular interactions with FNIII3 of the same protomer (Fig. 6c).

The FNIII2-FNIII2 interface was located at the center of the *cis* homodimer, where P395, P417 and Y420 formed a hydrophobic core (Fig. 6c). Y420 appeared to be essential because it formed a hydrogen bond with P395 and π–π interactions with the symmetry-related Y420. The FNIII1-FNIII2 interface was dominated by hydrogen bond networks formed by the side-chain of N357 and Q485, and the main-chain of L358 and E486 (Fig. 6c). In contrast, the FNIII1-FNIII3 interface was composed of both hydrophobic and hydrophilic interactions (Fig. 6c).

To verify the *cis*-dimer interfaces, we made single-site mutations on the FNIII1-FNIII2 interface (N357A and Q485A) and the FNIII2-FNIII2 interface (Y420A). Evaluated by the cKIT tyrosine phosphorylation assay, all the mutants significantly reduced the *cis*-interaction (Fig. 5c, e).

## Oligomerization status of sDscam Ig and FNIII domains in solution

To biochemically characterize the oligomerization status of sDscam Ig and FNIII domains in vitro, we compared the gel filtration profiles of different sDscam Ig or FNIII fragments (Supplementary Fig. 4). In the gel-filtration profile, the Ig1 peak was eluted earlier than the Ig2 or Ig3 peak, indicating that the Ig1 domain assembles into larger oligomers than Ig2 or Ig3 in solution. The FNIII domain has a similar molecular size and shape as those of the Ig domain. The single FNIII domains of sDscam were eluted at a similar position with that of Ig2 or Ig3, while FNIII2-3 was eluted at a position corresponding to the Ig1 homodimer. These data suggested that each single FNIII domain and the FNIII2-3 fragment existed as monomers in solution. The FNIII1-3 was eluted at a very similar position as that of Ig1-3, indicating the FNIII1-3 also existed as a homodimer in solution. The gel-filtration analysis supported our observations in the crystal structures: Ig1 mediated the formation of *trans* homodimer (Fig. 2) and all the three FNIII domains were involved in the *cis* interactions (Fig. 6).

## Model of the full extracellular domain assembly of sDscam

Neuronal self-avoidance requires a reliable discrimination between self and non-self neurites. Two encountering neurites from different cells should not incorrectly recognize each other as self. This would be simple when the two cells express completely different subsets of sDscam isoforms, since no homophilic binding would occur. However, there is a counterintuitively high probability of overlap of randomly expressed isoforms in different neurons, even with a very large repertoire of the isoform pool. For example, using Monte-Carlo simulations, it has been shown that, with 50 isoforms randomly selected per neuron from a pool of 20,000, the probability that two neurons express at least one common isoform is ~12%[12]. Therefore, some fraction of common isoforms between different neurons must be tolerated without triggering recognition.

A zipper model has been proposed for cPcdh intermembrane assembly[10], which provides a chain-termination mechanism to achieve an extremely high common-isoform tolerance. This explains how the much fewer cPcdh isoforms may be sufficient to mediate neuronal discrimination in vertebrates. Considering the comparable isoform number of cPcdh and sDscam, Chelicerata sDscam also needs a high tolerance for common isoforms and might adopt the zipper-like assembly as cPcdh.

If sDscam adopts the zipper-like assembly, our crystal structures of sDscam Ig and FNIII fragments covering the entire sDscam ectodomain enabled us to propose a model for the panorama of sDscam *trans* and *cis* recognitions. A distance of ~24 nm of the *Drosophila* Dscam1-mediated cell adhesion interface has been observed by electron microscopy[13]. We supposed that sDscam-mediated cell interface had a similar distance as that of *Drosophila* Dscam1, and the calculation arrived at 53° for the angle between sDscam Ig and FNIII domains (Fig. 7a).

Based on the crystal structures of *M. martensii* sDscam Ig1-3 *trans* dimer (isoform α25) and FNIII1-3 *cis* dimer (isoform β2v6), we proposed a molecular zipper model for sDscam recognition (Fig. 7a). On cell surface, two neighboring sDscam molecules form a cross *cis*-dimer via the FNIII domains in a side-by-side manner, while each extended arm (Ig domains) of a sDscam *cis*-dimer engages in *trans* with two different *cis*-dimers on an apposed cell in a hand-in-hand manner. Thus, a continuous array assembled through alternating *cis* and *trans* interactions of sDscam. In order to incorporate into the zipper-like assembly, the incoming sDscam *cis*-dimer must have one isoform to match the isoform on the exposed end of the zipper assembly (Fig. 7b). The zipper assembly will terminate when a matched isoform is not available, resembling the chain-termination model for cPcdh-mediated discrimination between self and non-self[10, 14].

## Discussion

In the nervous system, individualization and specialization of the cells are at the basis of their function, and thus a vast diversity of cell surface recognition proteins is a necessity. Alternative splicing of pre-messenger RNA is the most common mechanism to achieve the diversification[15]. To create diversified repertoires of recognition, different RNA splicing mechanisms have been evolved, such as mutually exclusive RNA splicing for fruit fly Dscam1[1] and alternative promoters for human cPcdh[4]. Chelicerata sDscam has been found to combine alternative promoters and RNA splicing to generate its isoform diversity[5], and therefore offers a unique perspective on the diversity and evolution of the neural cell surface receptors.

Our studies revealed two types of interactions of sDscam: the *trans* interactions and the *cis* interactions. The *trans* interactions of Chelicerata sDscam were mediated exclusively through the Ig1 domain for both sDscamα and sDscamβ subfamilies (Figs. 2 and 4). It is worth noting that sDscamβ subfamily has two variable Ig domains, Ig1 and Ig2. Although not involved in the *trans* interaction, the variable Ig2 domain may play other as-yet unknown functions, such as immune regulation observed for Arthropod Dscams[16,17]. The *cis* interactions are mediated by the FNIII domains that are constant within the sDscam subfamilies. We evaluated the *cis* polymerization of sDscam FNIII domains on living cell surface (Fig. 5) and determined the crystal structure of sDscam homo *cis*-dimer (Fig. 6). However, whether hetero *cis* interactions exit among sDscam subfamilies has not been evaluated due to the limitation of these assays. The *cis* interactions of cPcdh are promiscuous, but with preferences favoring formation of heterologous *cis*-dimers[7]. It has been reported that different sDscam subfamilies may form *cis*-dimers in co-IP assays[11]. Different isoforms within sDscamβ subfamily share a 41–61% identity of protein sequence for the FNIII part, while sDscamα and sDscamβ FNIII domains share a 35–42% protein sequence identity. Sequence alignment of these FNIII domains showed conservation of the interface residues, to a certain extent, among different subfamilies (Supplementary Fig. 5a). Whether hetero *cis*-dimers exit in sDscam and if so, whether they adopt the same conformation as the homo *cis*-dimer need further studies. Sequence analysis of all the FNIII domains of *M. martensii* sDscams showed that all the potential glycosylation sites located at the middle of FNIII1 β-strand B and the beginning of FNIII2 β-strand E (Supplementary Fig. 5a). The two locations are both exposed to the solution and the glycosylation would not interfere with the *cis*-dimer interfaces observed in the crystal structure (Supplementary Fig. 5b).

The isoform diversity varies enormously in insect Dscam and vertebrate cPcdh, and therefore two different strategies have been used for cell recognition. *Drosophila* Dscam1 with a vast isoform repertoire (38,016 isoforms) appears to perform *trans* interactions as monomers, while human cPcdh with a limited isoform diversity (52 isoforms) forms *cis* multimers to further assemble a zipper-like structure between cell membranes[10,18,19]. The cryo-electron tomography revealed that cPcdh forms zipper-like assemblies in a native-mimicking membrane environment[19]. However, such continuous ordered assemblies have not been observed for *Drosophila* Dscam1 on cell surface[13]. To avoid incorrect recognition between non-self neurites, some fraction of common isoforms between different neurons must be tolerated without triggering recognition[12]. It has been demonstrated that thousands of isoforms are necessary for Dscam1, with an ~20% tolerance, to provide enough number of neurons with functionally unique identities[12]. Compared to the thousands of isoforms of Dscam1, clustered Pcdh only has tens of isoforms and thus a higher tolerance is required. This zipper model provides a chain-termination mechanism: even a single mismatch is sufficient to disrupt the chain extension and thus to significantly reduce the number of favorable interactions. Therefore, an extremely high common-isoform tolerance is achieved.

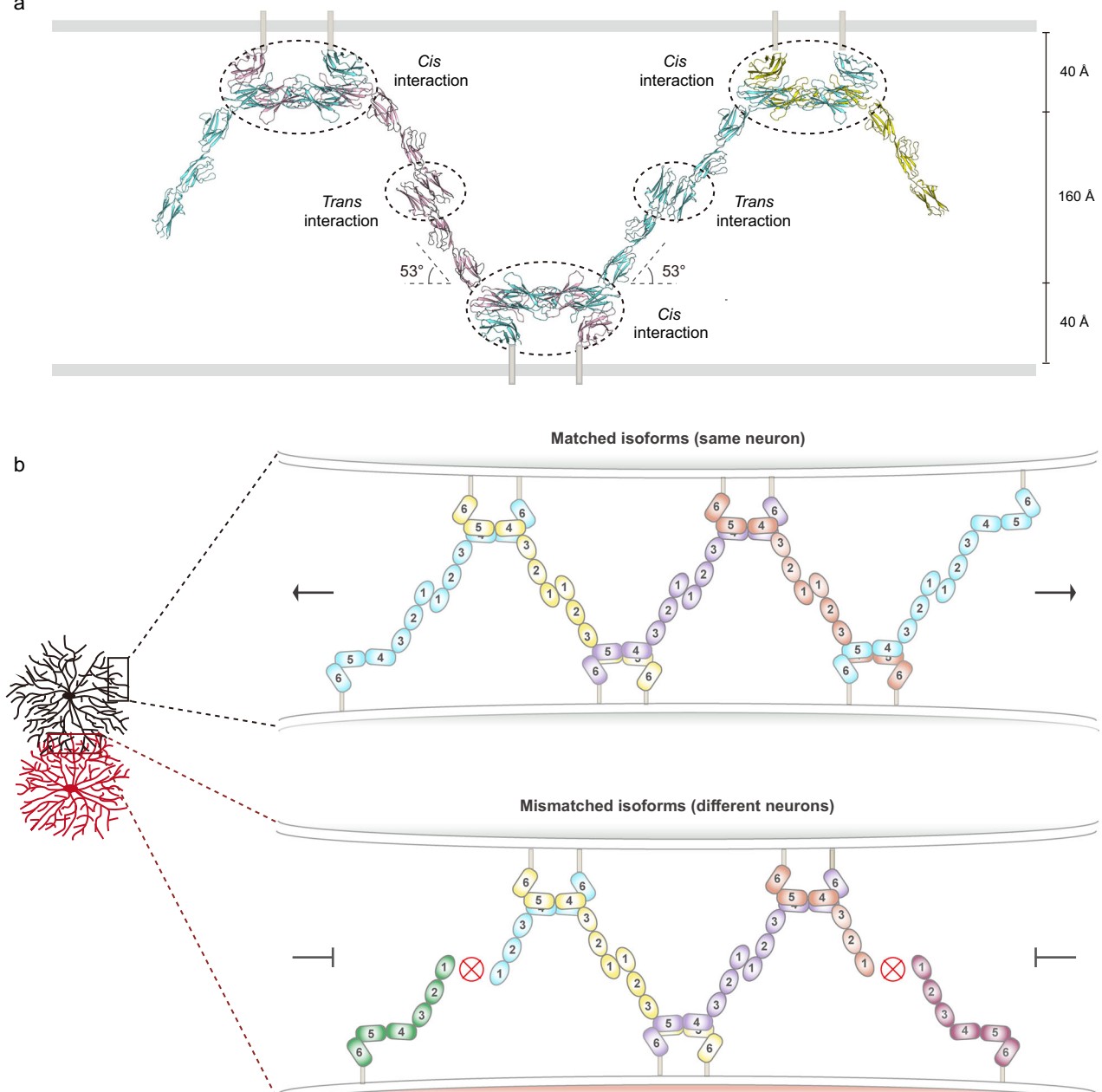

**Fig. 7 | The zipper-like model for sDscam recognition. a** Schematic of the zipper-like structure of sDscam molecules between apposed membrane surfaces. The domains involved in the *cis* or *trans* interactions are indicated by dashed circles. **b** Proposed zipper-like model for sDscam recognition. Chain extension of sDscam molecules occurs between the dendrites of the same neuron which have the same isoform repertoire, resulting in the sDscam-mediated cell-cell recognition. A switch-like repulsion response might be elicited when the extension of the zipper reaches a threshold. On the contrary, dendrites of different neurons owning mismatching isoforms will terminate chain extension.

Although Chelicerata sDscam shared a high sequence homology to invertebrate Dscam and used Ig1 domain to mediate *trans* interaction mimicking Dscam1 Ig7 domain (Fig. 2), its limited isoform diversity implied that it might use the zipper-like assembly strategy for cell recognition as vertebrate clustered Pcdh. The *trans*-interaction of the membrane-distal Ig1 domain and the *cis*-interaction of the membrane-proximal FNIII domains observed in the crystal structures of sDscam displayed the characteristics of the zipper formation.

However, many sDscamβ isoforms of *M. martensii* could not form homophilic aggregates[11]. If these sDscamβ isoforms are unable to form *trans* interactions, the incorporation of them into the zipper would result in chain termination even when all isoforms match. Failure to form cell aggregates has also been observed in mammalian Pcdhα

isoforms which is due to a lack of membrane localization[18,20,21]. The situation of sDscam seems more complicated, since at least one beta sDscam, β4v3, does localize to the membrane surface[11]. While the wild-type sDscam β4v1 could not form cell aggregates, deletion of its FNIII domain or replacing its Ig3 or FNIII1 domain with the counterpart of α14 endowed this isoform with the ability of forming cell aggregates[11]. In turn, the replacement of sDscam α14 Ig3 domain with the counterpart of β4v1 abolished α14's ability of forming cell aggregates[11]. Since the *trans*-interaction of sDscam is mediated by Ig1, how other domains affect the *trans*-interaction remains elusive. The failure to form cell aggregates has only been observed in sDscamβ family, while all sDscamα isoforms exhibited extensive aggregation in the cell aggregation assays[11]. It is likely that some unknown mechanism regulates the

isoform-specific *trans* interactions of sDscamβ isoforms. So, either some sDscamβ isoforms do form *trans* dimers but fail to form cell aggregates for unknown reasons, or sDscam has a unique self-recognition strategy other than the zipper model.

Arthropoda, the largest phylum in the animal kingdom, accounts for more than 80% of known animal species. Chelicerata is the second largest subphylum of arthropods, the evolutionary history of which could extend back to the Cambrian (~524 million years ago)[22,23]. The origin of s*Dscam* genes has been proposed to lie before the split of Arachnida and Merostomata, and phylogenetic analysis suggests that the sDscam may originate from the sequential shortening of the canonical Dscam[5,6]. Dscam1 has been implicated as a Netrin-1 receptor and the Netrin-1 binding site is located at the Ig7-Ig9 fragment[24,25]. Interestingly, the three Ig domains of Chelicerata sDscam correspond to Dscam1 Ig7-Ig9[5], implying that the shortened sDscam keeps the Netrin-1 binding site in evolution and plays a similar role as Dscam1 in neural development.

Diverged from other Arthropod lineages ~500 million years ago, Chelicerata represented the insect Dscam homolog, sDscam, which has a genomic organization similar to vertebrate clustered Pcdh. Our structural and functional results reported here showed that sDscam Ig1 mediated *trans* interaction resembling insect Dscam1 Ig7, while the membrane-proximal FNIII domain mediated *cis* interaction and might form a zipper-like assembly mimicking the vertebrate clustered Pcdh. Together, these studies advanced our understanding of sDscam and shed light on the evolutionary landscape of the recognition molecule diversity.

## Methods

### Protein production and crystallization

The sDscam Ig1-2, Ig1-3 constructs and *M. martensii* isoform α7 were cloned into a vector derived from pVL1392 (SnapGene) and expressed in High5 insect cells using a baculovirus system (BacMagic, Novagen). Other Ig fragments and all the FNIII fragments were cloned into a vector derived from pET-28a (+) (Novagen), which contains a TEV protease cleavage site after the N-terminal His6 tag. The Ig fragments were expressed in *E. coli* strain SHuffle (NEB), while the FNIII fragments were expressed in *E. coli* strain Rosetta (DE3) (Novagen). The protein purification was performed as previously described[26]. Briefly, the cells were harvested by centrifugation and resuspended in binding buffer (20 mM Tris−HCl pH 7.4, 250 mM NaCl) before lysis. Then the target proteins were purified by IDA-Nickel magnetic beads (BeaverBeads™, Beaverbio), followed by tag-removing using TEV protease (1:100). The digests were first passed through a desalting column to remove imidazole and then a Ni−NTA column (GE Healthcare) to remove free His6 tag, uncleaved protein and TEV protease. Target proteins in the flow-through were collected and further purified via gel filtration (Superdex 10/300 GL, GE Healthcare) in a buffer consisting of 20 mM HEPES, 100 mM NaCl pH 7.5. Crystals were grown at 16 °C using the hanging-drop vapor diffusion method. The crystallization conditions were summarized in Supplementary Table 1. The primer sequences were provided in Excel format as separate Supplementary Data 1.

### Data collection and structure determination

Diffraction data were collected on beamline BL19U1 at the Shanghai Synchrotron Radiation Facility (SSRF). Collected data were processed by HKL3000[27]. The crystal structures of single Ig or FNIII domains were determined by molecular replacement using the predicted structures from AlphaFold[28] as search models. The multidomain structures were determined by molecular replacement using the structures of single domains as search models. Structure refinement and model building were performed with PHENIX[29] and Coot[30]. All models were validated

with MolProbity[31]. Details of the data processing and refinement statistics were summarized in Supplementary Table 2. All structure figures were prepared with ChimeraX[32] and PyMOL (https://www.pymol.org).

### Cell aggregation assay

Cell aggregation assay was performed as previously reported with minor modifications[11]. Sf9 cells were cultured in SIM SF medium (Sino Biological) supplemented with 10% fetal bovine serum (Gibco) at 27 °C. Baculoviruses were obtained according to the manufacturer's instructions of BacMagic transfection kit (Novagen). Sf9 cells were seeded at 8x10^5 cells per well in a 6-well plate and infected with P2 recombinant viruses followed by 2-day incubation at 27 °C. Then cells were collected by centrifuge, washed once by 1 × HCMF (Coolaber) gently and resuspended with 1 × HCMF. For cell aggregation assay, the 6-well plate preincubated by 1% BSA in 1 × HBSS (Solarbio) at 4 °C overnight and washed with 1 × HCMF. Then 2 × 10^5 cells in 2 ml 1 × HCMF were transferred into each well of the 6-well plate. Cell suspension in 6-well plates was incubated at 27 °C in gyratory shaker at 60 rpm for 1 h. Samples were then observed with inverted fluorescence microscope Ti-S (Nikon).

### Quantification of the size of cell aggregates

The quantitative analysis of cell aggregates was carried out by MATLAB according to the previous study[11]. The images were converted to the black-and-white format with 2160 × 2560 pixels. The objects with 300 or fewer pixels (smaller than 3 cells) were classified as "no aggregation", while objects with more than 300 pixels (larger than 3 cells) were classified as "aggregation". Then, the sum of object pixels larger than 300 pixels was divided by the sum of all object pixels to calculate the percentage of cell aggregation.

### Calculation of coaggregation index

Coaggregation (CoAg) was calculated by MATLAB in accordance with the previous studies[11,33]. Briefly, each image of cell aggregation was parsed into squares just slightly larger than the size of a single cell. Then, all black squares (empty area containing no cells) were removed from the image, and the remaining squares were used for the analysis (the percent of squares that contain more than one color was calculated). As a result, the completely separated cells would have a very low CoAg index (<0.1), while the intermixing cells would have a high CoAg index (≥0.2).

### Tyrosine phosphorylation assay

Sf9 cells were cultured in SIM SF medium (Sino Biological) supplemented with 10% fetal bovine serum (Gibco) at 27 °C. For infection, cells were seeded at 2x10^6 cells per well in a 6-well plate, infected with P2 recombinant viruses followed by 3-day incubation at 27 °C. Cells were then collected by centrifuge, washed twice with PBS, solubilized in lysis buffer containing 120 mM NaCl, 25 mM HEPES (pH 7.4), 1 mM EGTA, 0.75 mM MgCl$_2$, 10% glycerin, 1% triton, 1 mM NaF, 2 mM sodium orthovanadate, protease inhibitor cocktail (EDTA-free), and incubated on ice for 1 h. The lysates were centrifuged at 13,400×*g* for 20 min at 4 °C and then the supernatants were incubated with anti-Flag magnetic beads (Smart Lifesciences,) for 2 h at 4 °C with rotating. The beads were washed twice with washing buffer containing 50 mM HEPES (pH 7.4), 150 mM NaCl, 1 mM EGTA, 0.75 mM MgCl$_2$, 10% glycerol, 0.1% triton, and 2 mM sodium orthovanadate. Finally, the samples were boiled for 5 min and analyzed by western blot with anti-Flag (Smart Lifesciences, Cat. No. SLAB01, 1:5,000 dilution) and anti-phosphotyrosine (HuaBio, Cat. No. ET1704-20, Clone No. JA10-49, 1:100 dilution) antibodies. Protein was visualized by chemiluminescence imager ChemiDoc Touch (Bio-Rad) and the quantifications were performed using ImageJ program.

## Statistical analysis

Statistical analysis was performed using the GraphPad Prism 8 Software. Two-tailed Student's $t$ test was used to compare differences. Significance level was set at $P > 0.05$. All values were reported as means ± s.d.

## Reporting summary

Further information on research design is available in the Nature Portfolio Reporting Summary linked to this article.

## Data availability

Atomic coordinates and structure factors in this study have been deposited in the Protein Data Bank (PDB) under accession codes: 7Y54 (α1-Ig1), 7Y4X (α7-Ig1), 7Y9A (β2v6-Ig1-2), 7Y95 (β6v2-Ig1), 7Y6O (α25-Ig1-3), 7Y5J (α1v7-Ig1), 7Y73 (β3v7-Ig1), 7Y8H (α7-FNIII1), 7Y5R (α7-FNIII2), 7Y8I (α7-FNIII3), 7Y6E (β2v6-FNIII2-3), and 7Y8S (β2v6-FNIII1-3). Source data are provided with this paper.

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

## Acknowledgements

We thank the staffs from BL19U1 beamline of National Center for Protein Science Shanghai (NCPSS) at Shanghai Synchrotron Radiation Facility for assistance during data collection. Financial support for this work was provided by National Natural Science Foundation of China (32270761 to Q.C., 31741027 to Y.Y., and 31630089 to Y.J.), 1·3·5 Project for Disciplines of Excellence, West China Hospital, Sichuan University (ZYJC21073 to X.Z.), and the Starry Night Science Fund at Shanghai Institute for Advanced Study of Zhejiang University (SN-ZJU-SIAS-009 to Y.J.).

## Author contributions

Q.C. conceived and designed the experiments. J.C., X.W., and T.L. performed protein preparation and crystallization. Y.Y and Q.C. collected diffraction data, solved the structures, and conducted the structural analysis. J.C. and D.H. performed cell aggregation assay and Tyrosine phosphorylation assay. X.Z. performed statistical analysis. Q.C. wrote the manuscript with the comments from T.-M.F., Y.L., and Y.J.

## Competing interests

The authors declare no competing interests.
