## [Peer Review File · Nature Communications]

REVIEWER COMMENTS

Reviewer #1 (Remarks to the Author):

In this paper the authors describe the structures of cis homodimers as well as trans homodimers for several sDSCAM isoforms. The structures are quite interesting as is the speculation that these proteins use a similar mechanism to that used by protocadherins (cPcdh) for neuronal barcoding. However, the connection is not as clear-cut as the authors assume.

The authors also add a mathematical model analysis similar to the one reported in Thu et al. 2014 to demonstrate that, like cPcdh, sDscam also provides sufficient diversity for the neuronal self-avoidance process. The main problem with this paper is that the authors present sDscam mediated neuronal self-avoidance as a given, and this is misleading. For example, at the end of the second paragraph in the introduction after discussing neuronal self-avoidance it says: "How this shortened sDscam fulfills its functional roles remains elusive..." thereby seemingly indicating that we know the function of sDscam and all that is left is to now understand the mechanism by which this function is carried out. The reality is that the biological function of sDscam remains unknown.

In addition, the authors propose a molecular zipper model for sDscam recognition that could provide sufficient cell surface diversity for self recognition. This model is identical to the chain termination model proposed for cPcdhs, where even a single mismatch is sufficient to disrupt chain extension. However, the application of this model to sDscam is problematic. As shown in a recent publication, almost 40 of the 95 sDscam isoforms are unable to form trans interactions as tested in cell aggregation assays (Zhou et al PNAS 2020). Once any of these isoforms is incorporated into the zipper, the result would be chain termination even when all isoforms match.

I believe the authors need to discuss these issues and provide a thoughtful discussion based on the known biology.

Reviewer #2 (Remarks to the Author):

The manuscript entitled "Structural Basis for the self-recognition of sDSCAM in Chelicerata" fills an important gap in the understanding of the role of the variable exons of Dscam in generating a set of cell

surface receptors with distinct adhesion properties that regulate self-recognition between neurons. Down syndrome cell adhesion molecule (DSCAM) has been the focus of intense research, to better understand the molecular mechanisms underlying neuronal organization. This study provides a detailed analysis of an interesting intermediate form of Dscam that is only observed in the subspecies of Chelicerata, which have a reduced set of Dscam isoforms that still contribute to self recognition between neurons. This manuscript provides an important contribution to the understanding of the molecular mechanisms underlying neuronal wiring, and is of general interest to neurobiologists and biophysicists interested in molecular organization principles. However, there are several issues that should be addressed before this paper could be considered for publication:

1. The cis-dimer identification experiment that uses sDscam-cKIT chimeras is a nice tool to identify which sDscam domains contribute to dimerization on the same cell surface. The authors should however complete their analysis and add chimeras that contain the Ig123 domains, as well as the full length ectodomain, to show that cis dimerization is indeed only caused by the fibronectin domains. They should also include mutants to verify the cis dimer interface they identify (for instance a Y420R mutant).
2. The fibronectin domain constructs that were crystallized were produced in *E. coli*, which lacks a glycosylation machinery. The lack of glycosylation may alter the dimer interfaces. Could the authors comment on the possible interference of glycosylation in their proposed cis-dimer interface? I also would like to see a more detailed analysis of the buried surface area of the fibronectin dimer interfaces, and would like to analyze the PDB coordinates to verify this analysis.
3. The conclusions drawn from cell aggregation assays used to identify isoform specific homophilic interactions are based on single images of cell aggregates. The authors should provide a more robust statistical analysis of these aggregation assays, for instance by quantifying the number of cells involved in homo or heterophilic aggregates over a certain area.
4. The authors should better describe the diversity of the protocadherin family, which consists of alpha, beta and gamma protocadherins. They cite the original PCDH paper from 1999, but they should also check more recent literature. See for instance <https://elifesciences.org/articles/72416>, which mentions about 60 PCDH isoforms instead of 52 mentioned in this manuscript.
5. That paper also describes cross-reactivity for cis interactions in PCDHs, which is relevant for the zipper-like model proposed in the current paper as well. Figure 7b & c shows sDscam ectodomains with different colors, which is confusing, because I assume the authors want to suggest that zippers form when identical isoforms engage. In 7b there is a green ectodomain forming a trans dimer, whereas in 7c this green ectodomain cannot engage.

6. The mathematical modeling for the tolerance for non-self recognition depends on so many assumptions that it is almost arbitrary. 1) It assumes that there are 10 isoforms per neuron, which is not based on any experimental data (in fact Neves et al suggest there are 14-50 distinct isoforms per neuron for *Drosophila* Dscam). 2) It assumes that sDscam has a similar tolerance level as PCDH, although it has more distinct isoforms, and 3) the tolerance of 80% for PCDHs is also an assumption itself. It is therefore not clear to me what this mathematical analysis shows, other than that this is a very complex system. I would remove the mathematical modeling from the paper.

REVIEWER COMMENTS

Reviewer #1 (Remarks to the Author):

In this paper the authors describe the structures of cis homodimers as well as trans homodimers for several sDSCAM isoforms. The structures are quite interesting as is the speculation that these proteins use a similar mechanism to that used by protocadherins (cPcdh) for neuronal barcoding. However, the connection is not as clear-cut as the authors assume.

We thank the reviewer for the positive comments.

The authors also add a mathematical model analysis similar to the one reported in Thu et al. 2014 to demonstrate that, like cPcdh, sDscam also provides sufficient diversity for the neuronal self-avoidance process. The main problem with this paper is that the authors present sDscam mediated neuronal self-avoidance as a given, and this is misleading. For example, at the end of the second paragraph in the introduction after discussing neuronal self-avoidance it says: "How this shortened sDscam fulfills its functional roles remains elusive..." thereby seemingly indicating that we know the function of sDscam and all that is left is to now understand the mechanism by which this function is carried out. The reality is that the biological function of sDscam remains unknown.

We thank the reviewer for pointing out this issue. There are no reports on the biological functions of sDscam, which is probably due to the lack of a model organism in Chelicerata.

The generation of a large number of isoforms and the isoform-specific interaction are two hallmarks of Dscam/cPcdh mediated neuron recognition and self-avoidance. Chelicerata sDscam generates ~50 to 100 isoforms via alternative promoter and RNA splicing selections, and exhibited isoform-specific trans homophilic interactions. So, it is rational to speculate that sDscam plays a similar function as Dscam and cPcdh.

We have revised our statement on this issue in the introduction to make it clear that sDscam playing a similar function as Dscam/cPcdh is a speculation.

In addition, the authors propose a molecular zipper model for sDscam recognition that could provide sufficient cell surface diversity for self recognition. This model is identical to the chain termination model proposed for cPcdhs, where even a single mismatch is sufficient to disrupt chain extension. However, the application of this model to sDscam is problematic. As shown in a recent publication, almost 40 of the 95 sDscam isoforms are unable to form trans

interactions as tested in cell aggregation assays (Zhou et al PNAS 2020). Once any of these isoforms is incorporated into the zipper, the result would be chain termination even when all isoforms match.

I believe the authors need to discuss these issues and provide a thoughtful discussion based on the known biology.

We agree with the reviewer that the negative results of *trans*-interaction of some sDscam isoforms are incompatible with the molecular zipper model.

While all sDscam α isoforms exhibited extensive aggregation in the cell aggregation assays, many sDscam β isoforms could not form homophilic aggregates (Zhou *et al.* PNAS 2020, Fig. 1D). Failure to form cell aggregates has also been observed in mammalian Pcdh α isoforms which is due to a lack of membrane localization (Thu et al., Cell 2014; Bonn et al., Mol. Cell. Biol. 2007; Murata et al., J. Biol. Chem. 2004).

The situation of sDscam seems more complicated. For example, while the wild-type sDscam β 4v1 could not form cell aggregates, deletion of its FNIII domain or replacing its Ig3 or FNIII1 domain with the counterpart of α 14 endowed this isoform with the ability of forming cell aggregates (Zhou *et al.* PNAS 2020, Fig. S2C, S2E & S3F). In turn, the replacement of sDscam α 14 Ig3 domain with the counterpart of β 4v1 abolished α 14's ability of forming cell aggregates (Zhou *et al.* PNAS 2020, Fig. S2E & S3F). We have shown that the *trans*-interaction of sDscam is mediated by Ig1. How other domains effect the *trans*-interaction remains elusive.

The failure to form cell aggregates is not random among sDscam isoforms. Instead, it seems to be isoform-specific (only in sDscam β family). So, it is likely that some unknown mechanism regulates the isoform-specific *trans* interactions of sDscam β isoforms.

We have added a discussion on this issue according to the reviewer's suggestion.

Reviewer #2 (Remarks to the Author):

The manuscript entitled "Structural Basis for the self-recognition of sDSCAM in Chelicerata" fills an important gap in the understanding of the role of the variable exons of Dscam in generating a set of cell surface receptors with distinct adhesion properties that regulate self-recognition between neurons. Down syndrome cell adhesion molecule (DSCAM) has been the focus of intense research, to better understand the molecular mechanisms underlying neuronal organization. This study provides a detailed analysis of an interesting

intermediate form of Dscam that is only observed in the subspecies of Chelicerata, which have a reduced set of Dscam isoforms that still contribute to self recognition between neurons. This manuscript provides an important contribution to the understanding of the molecular mechanisms underlying neuronal wiring, and is of general interest to neurobiologists and biophysicists interested in molecular organization principles. However, there are several issues that should be addressed before this paper could be considered for publication:

We thank the reviewer for the positive comments.

1. The cis-dimer identification experiment that uses sDscam-cKIT chimeras is a nice tool to identify which sDscam domains contribute to dimerization on the same cell surface. The authors should however complete their analysis and add chimeras that contain the Ig123 domains, as well as the full length ectodomain, to show that cis dimerization is indeed only caused by the fibronectin domains. They should also include mutants to verify the cis dimer interface they identify (for instance a Y420R mutant).

We highly appreciate the positive comments on our living cell-based assay.

The full length ectodomain containing Ig domains will involve the *trans* interaction and cause cell aggregation. This may make this problem complexed. We want to exclude the *trans* interaction and study the *cis* interaction in the absence of cell aggregation. Since the Ig domains along could also cause cell aggregation (Zhou *et al.* PNAS 2020, Fig. S2C), we have not used the Ig123 domains as a control. Crystal structure of different fragments of the sDscam Ig domains supported that they mediated *trans* but not *cis* interaction.

According to the reviewer's suggestion, we made mutants to verify the cis dimer interface: N357A and Q485A for the FNIII1-FNIII2 interface, Y420A for the FNIII2-FNIII2 interface. Y420 formed a hydrogen bond with P395 and π - π interactions with the symmetry-related Y420. We do not use the Y420R mutant because arginine might make hydrogen bonds via its guanidinium moiety and make hydrophobic interactions via its long hydrophobic stalk. We have not made mutations in the FNIII1-FNIII3 interface, since it involved too many residues and seemed hard to be abolished by single-site mutations.

The mutants N357A, Q485A and Y420A significantly reduced the *cis*-interaction (Fig. 5c & 5e), which supported our crystal structure of sDscam *cis*-dimer.

2. The fibronectin domain constructs that were crystallized were produced in E.

coli, which lacks a glycosylation machinery. The lack of glycosylation may alter the dimer interfaces. Could the authors comment on the possible interference of glycosylation in their proposed cis-dimer interface? I also would like to see a more detailed analysis of the buried surface area of the fibronectin dimer interfaces, and would like to analyze the PDB coordinates to verify this analysis.

We thank the reviewer for pointing out this issue. Sequence analysis of all the FNIII domains of *M. martensii* sDscams showed that all the potential glycosylation sites located at the middle of FNIII1 β -strand B and the beginning of FNIII2 β -strand E (Fig. S6a). The two locations are both exposed to the solution and the glycosylation will not interfere with the *cis*-dimer interfaces observed in the crystal structure (Fig. S6b).

We provide the PDB file of FNIII1-3 structure for the verification.

3. The conclusions drawn from cell aggregation assays used to identify isoform specific homophilic interactions are based on single images of cell aggregates. The authors should provide a more robust statistical analysis of these aggregation assays, for instance by quantifying the number of cells involved in homo or heterophilic aggregates over a certain area.

We appreciate the reviewer's suggestion. In this revision, we have provided a statistical analysis of these aggregation assays (Fig. 5).

4. The authors should better describe the diversity of the protocadherin family, which consists of alpha, beta and gamma protocadherins. They cite the original PCDH paper from 1999, but they should also check more recent literature. See for instance <https://elifesciences.org/articles/72416>, which mentions about 60 PCDH isoforms instead of 52 mentioned in this manuscript.

According to the reviewer's suggestion, we described the diversity of the cPcdh family in the Introduction part of the revision and cited the recent eLife paper as ref. 7.

The 52 isoforms we have mentioned in the Introduction part are for human cPcdh family, while the ~60 cPcdh isoforms are for mouse. We used mouse cPcdh for the mathematical modeling (Fig. 7a & S5).

5. That paper also describes cross-reactivity for cis interactions in PCDHs, which is relevant for the zipper-like model proposed in the current paper as well. Figure 7b & c shows sDscam ectodomains with different colors, which is confusing, because I assume the authors want to suggest that zippers form when identical isoforms engage. In 7b there is a green ectodomain forming a trans dimer, whereas in 7c this green ectodomain cannot engage.

Yes, we want to show that identical isoforms engage to form zippers. In this revision, we have changed the colors in Fig. 7b to ensure a same color for the identical isoforms.

6. The mathematical modeling for the tolerance for non-self recognition depends on so many assumptions that it is almost arbitrary. 1) It assumes that there are 10 isoforms per neuron, which is not based on any experimental data (in fact Neves et al suggest there are 14-50 distinct isoforms per neuron for *Drosophila Dscam*). 2) It assumes that sDscam has a similar tolerance level as PCDH, although it has more distinct isoforms, and 3) the tolerance of 80% for PCDHs is also an assumption itself. It is therefore not clear to me what this mathematical analysis shows, other than that this is a very complex system. I would remove the mathematical modeling from the paper.

We thank the reviewer for pointing out this issue.

There are 14-50 distinct isoforms per neuron for *Drosophila Dscam* (Neves et al., *Nat Genet* 2004) while the number is 10~15 for *Pcdh* (Mountoufaris et al., *Annu Rev Cell Dev Biol* 2018). In this revision, we calculated the probabilities that a pair of neurons share common isoforms assuming that there are 10, 15, 20, 30, 40, and 50 isoforms per neuron, respectively (Fig. S5). For the various numbers of the sDscam isoforms expressed per cell, the common-isoform tolerance would be 70~80% to obtain a low enough probability of incorrect non-self recognition (for example, 1×10^{-8}).

It has been shown that even four of five expressed *Pcdh* isoforms being identical could avoid incorrect non-self recognition, implying a common-isoform tolerance of at least 80% (Thu et al., *Cell* 2014). To have a better comparison, we used a common-isoform tolerance of 80% for both *Pcdh* and sDscam to calculate the probabilities of incorrect non-self recognition (Fig. 7a).

The mathematical modeling explains why sDscam needs a high common-isoform tolerance to avoid incorrect non-self recognition (Fig. S5), and with a given high common-isoform tolerance (80%, Fig. 7a), how much distinct isoforms are needed to be expressed on a single neuron cell. The mathematical calculations support our speculation that sDscam adopts the zipper-like assembly as clustered *Pcdh*. So, we tend to keep the mathematical modeling in this paper.

REVIEWER COMMENTS

Reviewer #1 (Remarks to the Author):

My main objection to the manuscript remains unchanged – the authors attempt to relate their results to previous work on fly Dscams and clustered protocadherins (cPcdh) but fail to do so. They have toned down the introduction to say that sDscams are likely to play a similar role to cPcdh and fly DSCAM, rather than stating that this is a known fact. However, in the discussion section, the authors fail to offer a convincing argument regarding the compatibility of sDscam with the zipper model for neuronal self recognition that has been proposed for cPcdhs. . A major issue is that, as opposed to the strict homophilic binding of cPcdhs, many beta sDscam isoforms do not form homophilic interactions in their assay. They need to explain (as I previously suggested) how these observations impact compatibility with the zipper model. They mention the alpha cPcdh's failure to form cell aggregates on their own but this is due to their failure to reach the cell surface without a "carrier" Pcdh. This is not an issue for beta sDscams that do localize to the membrane surface. The authors seem to assume that there are conditions where beta sDscams will form homophilic interactions but they don't know what they are. This assumption is again evident in the mathematical model in figure 7a (a repeat of the calculations done by Thu et al.). Here, the authors use all 95 sDscam isoforms as the total isoform pool available even though for almost half of these isoforms, they cannot find conditions where they form homophilic interactions.

I also agree with the second reviewer that the mathematical model in figure 7a is unnecessary and distracting. Based on the work done for cPcdhs and the number of isoforms available for sDscam the authors could conclude that sDscam would need a high tolerance for common isoforms without the model. If they suggest a zipper model, they would still need to discuss the issues that the beta sDscams present to this model.

In summary, the interesting structural results are diminished in this manuscript with unproven speculations as to the function of the molecules in question.

Reviewer #2 (Remarks to the Author):

The authors have addressed a number of issues raised by the two reviewers, but there are still some important points that need to be addressed:

1. This reviewer asked for coordinates and structure factors of the relevant structures, and one such data set was provided. Analysis of this data set using the PDB_REDO server showed that this one structure has not been refined fully yet. PDB_REDO reports better R factors (RfREDO= 21.5%/original file 22.6%;RfreeREDO = 27.4%/28.4%), better stereochemistry and also an analysis of the dimer interface with the PISA server reveals that the PDB_REDO structure has twice the number of hydrogen bonds in the dimer interface than the submitted coordinates reported in the manuscript. This means that the cis-dimer interface has to be reanalyzed and potentially Figures 6b and 6c will have to be remade.

2. Given the state of the one structural data set that was provided, this reviewer would like to see all other 11 data sets. I also recommend that the authors at least consider further refinement after analysis of the PDB_REDO output.

3. Both reviewers raise similar issues regarding the mathematical modelling, where the authors make many assumptions that are either arbitrary or contradict previous observations (for instance that a large number of sDscam isoforms do not form productive trans-dimers). But let's for a moment assume that the modelling is done correctly. What the model presented by the authors shows is that sDscam shows an anomaly with much lower chance for chain termination for the most probable range of isoforms per cell (10-20 isoforms). So this either indicates that sDscam requires other cell surface receptor systems to compensate its self-recognition function, or Chelicerata have a unique self-recognition system and an aberrant neuronal network.

So I insist that this modelling is removed from the paper, because a) the assumptions made are arbitrary and b) their modelling suggests there are species-specific neuronal self-recognition anomalies.

4. The authors should be more graceful in citing the literature on Protocadherins and Dscams. The original structural paper on Dscam that shows how isoforms recognize self through palindromic recognition sequences should be cited for instance (Meijers et al. Nature 2007).

REVIEWER COMMENTS

We highly appreciate the reviewers for their comments that have significantly improved our manuscript. We have carefully revised our manuscript according to the reviewers' suggestions.

Reviewer #1 (Remarks to the Author):

My main objection to the manuscript remains unchanged - the authors attempt to relate their results to previous work on fly Dscams and clustered protocadherins (cPcdh) but fail to do so. They have toned down the introduction to say that sDscams are likely to play a similar role to cPcdh and fly DSCAM, rather than stating that this is a known fact. However, in the discussion section, the authors fail to offer a convincing argument regarding the compatibility of sDscam with the zipper model for neuronal self recognition that has been proposed for cPcdhs. . A major issue is that, as opposed to the strict homophilic binding of cPcdhs, many beta sDscam isoforms do not form homophilic interactions in their assay. They need to explain (as I previously suggested) how these observations impact compatibility with the zipper model. They mention the alpha cPcdh' s failure to form cell aggregates on their own but this is due to their failure to reach the cell surface without a "carrier" Pcdh. This is not an issue for beta sDscams that do localize to the membrane surface. The authors seem to assume that there are conditions where beta sDscams will form homophilic interactions but they don' t know what they are. This assumption is again evident in the mathematical model in figure 7a (a repeat of the calculations done by Thu et al.). Here, the authors use all 95 sDscam isoforms as the total isoform pool available even though for almost half of these isoforms, they cannot find conditions where they form homophilic interactions.

We agree with the reviewer that more arguments regarding the compatibility of sDscam with the zipper model are needed. In this revision, we removed the mathematical model and carefully rewrote the discussion section to address more details of the sDscam β isoforms which could not form homophilic aggregates and its incompatibility with the zipper model.

I also agree with the second reviewer that the mathematical model in figure 7a is unnecessary and distracting. Based on the work done for cPcdhs and the number of isoforms available for sDscam the authors could conclude that sDscam would need a high tolerance for common isoforms without the model. If they suggest a zipper model, they would still need to discuss the issues that the beta sDscams present to this model.

According to the reviewer's suggestion, we have removed Figure 7a and Figure S5 and

the mathematical part from our manuscript.

We agree with the reviewer that based on the work on cPcdhs and the number of sDscam isoforms, sDscam would need a high tolerance for common isoforms. The zipper model provides a chain-termination mechanism which achieves an extremely high common-isoform tolerance for the high-tolerance requirement of cPcdh and sDscam.

From the structural perspective, the two modes of interactions observed in the sDscam crystal structures (the *trans*-interaction of the membrane-distal Ig1 domain and the *cis*-interaction of the membrane-proximal FNIII domains) are suitable for the formation of a zipper-like assembly.

Based on high-tolerance requirement and the *trans* and *cis* binding modes of sDscam, the zipper model seems a very suitable model for sDscam intermembrane assembly. We have modified our words to make it clear that it is only a possibility that sDscam adopts a zipper-like model, and provided a more detailed discussion on the beta sDscams issues in this revision.

In summary, the interesting structural results are diminished in this manuscript with unproven speculations as to the function of the molecules in question.

Reviewer #2 (Remarks to the Author):

The authors have addressed a number of issues raised by the two reviewers, but there are still some important points that need to be addressed:

1. This reviewer asked for coordinates and structure factors of the relevant structures, and one such data set was provided. Analysis of this data set using the PDB_REDO server showed that this one structure has not been refined fully yet. PDB_REDO reports better R factors (RfREDO= 21.5%/original file 22.6%;RfreeREDO = 27.4%/28.4%), better stereochemistry and also an analysis of the dimer interface with the PISA server reveals that the PDB_REDO structure has twice the number of hydrogen bonds in the dimer interface than the submitted coordinates reported in the manuscript. This means that the *cis*-dimer interface has to be reanalyzed and potentially Figures 6b and 6c will have to be remade.

We thank the reviewer for pointing out this issue. We further refined this FNIII1-3 structure based on the PDB_REDO output and accordingly revised Fig. 6 and Supplementary Fig. 5.

2. Given the state of the one structural data set that was provided, this reviewer would like to see all other 11 data sets. I also recommend that the authors at least consider further refinement after analysis of the PDB_REDO output.

According to the reviewer's suggestion, we have used PDB_REDO to check all the data sets of this manuscript. We found that for the relative-low-resolution structures ($\alpha 7$ -lg1, 2.95 Å; $\beta 2v6$ -lg1-2, 2.5 Å; $\alpha 25$ -lg1-3, 3.1 Å; $\beta 2v6$ -FNIII2-3, 3.05 Å; $\beta 2v6$ -FNIII1-3, 2.7 Å), PDB_REDO provided better results. For the high-resolution structures, PDB_REDO provided similar results ($\alpha 1$ -lg1, 1.8 Å; $\alpha 7$ -FNIII2, 1.56 Å) or even worse results ($\beta 6v2$ -lg1, 1.55 Å; $\beta 3v7$ -lg1, 1.32 Å). For some high-resolution structures ($\alpha 1v7$ -lg1, 1.6 Å; $\alpha 7$ -FNIII1, 1.8 Å; $\alpha 7$ -FNIII3, 1.9 Å), PDB_REDO provides a lower R value just because PDB_REDO uses Refmac to do refinement while we use PHENIX. Another run of PHENIX refinement of the PDB_REDO output of these structures gives similar or higher R values compared to our model. Thus, we further refined the five relative-low-resolution structures ($\alpha 7$ -lg1, 2.95 Å; $\beta 2v6$ -lg1-2, 2.5 Å; $\alpha 25$ -lg1-3, 3.1 Å; $\beta 2v6$ -FNIII2-3, 3.05 Å; $\beta 2v6$ -FNIII1-3, 2.7 Å) based on the PDB_REDO output and accordingly revised Fig.3 and Fig.6. We provided the mtz and pdb files for all the 12 data sets of this manuscript (structure_data.zip) with this revision.

PDB_REDO is very useful tool, especially for the relative-low-resolution structures, and we will routinely use this tool to polish our structures before submission.

3. Both reviewers raise similar issues regarding the mathematical modelling, where the authors make many assumptions that are either arbitrary or contradict previous observations (for instance that a large number of sDscam isoforms do not form productive trans-dimers). But let's for a moment assume that the modelling is done correctly. What the model presented by the authors shows is that sDscam shows an anomaly with much lower chance for chain termination for the most probable range of isoforms per cell (10-20 isoforms). So this either indicates that sDscam requires other cell surface receptor systems to compensate its self-recognition function, or Chelicerata have a unique self-recognition system and an aberrant neuronal network.

So I insist that this modelling is removed from the paper, because a) the assumptions made are arbitrary and b) their modelling suggests there are species-specific neuronal self-recognition anomalies.

According to the reviewer's suggestion, we have removed the mathematical part from our manuscript.

4. The authors should be more graceful in citing the literature on Protocadherins and Dscams. The original structural paper on Dscam that shows how isoforms recognize self through palindromic recognition sequences should be cited for instance (Meijers et al. Nature 2007).

We thank the reviewer for provide the information of the original paper. We have cited this paper (ref. 9) in this revision.

REVIEWERS' COMMENTS

Reviewer #2 (Remarks to the Author):

The authors have responded to all outstanding issues, and I congratulate them on a nice and thorough study. Please make sure that the updated PDB coordinates are shared with the PDB.

REVIEWERS' COMMENTS

Reviewer #2 (Remarks to the Author):

The authors have responded to all outstanding issues, and I congratulate them on a nice and thorough study. Please make sure that the updated PDB coordinates are shared with the PDB.

We thank the reviewer for the positive comments, and highly appreciate the help that has significantly improved our manuscript.

We have already deposited the updated coordinates in the Protein Data Bank (PDB), which will be released upon the publication of this manuscript.